# Challenges experienced by health care providers working in both hospital and home-based palliative care units in Dhaka city: A multi-center based cross-sectional study

**Mastura Kashmeeri**[1]*, **A. N. M. Shamsul Islam**[1], **Palash Chandra Banik**[2,3]

1 Department of Public Health & Hospital Administration, NIPSOM, Dhaka, Bangladesh, 2 Department of Noncommunicable Diseases, Bangladesh University of Health Sciences (BUHS), Dhaka, Bangladesh, 3 Center for Higher Studies and Research, Bangladesh University of Professionals, Dhaka, Bangladesh

* kshmeerimastura@gmail.com

**Data Availability Statement:** DOI: 10.17632/ykggdp85jw.1 https://data.mendeley.com/datasets/ykggdp85jw/1.

## Abstract

### Background

Palliative care is paramount in the modern clinical field worldwide. However, in Bangladesh, its acceptance is limited compared to other related sectors, despite the country suffering from a huge burden of life-limiting diseases. Besides, PC teams and their approach to care are entirely different from the conventional clinical approach. This study aimed to explore the challenges faced by healthcare providers working in the palliative care unit in Bangladesh, including all groups.

### Design

This was a cross-sectional descriptive survey involving palliative care providers.

### Methods

A self-administered pre-tested questionnaire was used for data collection. Data was analyzed using descriptive statistics and Chi-square at p <0.05.

### Result

The mean age of the respondents was 33.59 ± 8.05 years, and barely most (82.5%) had served for 7–9 years. More than half (51%) of doctors and 31% of nurses claimed patient agitation as a challenge. Almost all groups of respondents exhibit ethical dilemma as a barrier, although a significant relationship was found between professional level and ethical dilemma. More than half of doctors (51%), 41.5% of nurses, and 29.5% of PCA-ward staff mentioned the lack of telemedicine facilities as a challenge. Nearly half (47.1%) of doctors and nurses claimed that patients' families had made patient care difficult, on the other hand, PCA-ward staff (70%) group ignorance of family did the same thing. Opioid phobia of other health professionals restricted the growth mentioned by the majority of all four groups of respondents. A significant relationship was found between limited dose formulation and

**Funding:** The authors received financial support from the Bangladesh Medical Research Council (BMRC). However, the funders had no involvement in the study design, data collection and analysis, the decision to publish, or the preparation of the manuscript.

**Competing interests:** The authors declare that there is no conflict of interest.

experience of HPs (p<0.07). At the institutional level, 93.3% of nursing staff agreed that the lack of supporting staff was a drawback. A significant relationship was also found between the type of institution and the lack of a support system to conduct home-based care (p<0.002). Moreover, the majority (83.3%) of PCA-WS exhibit a lack of career development opportunities (p<0.001) as a barrier, besides, more than 7 out of 10 doctors (7.2%) felt social discrimination as a challenge(p<0.001).

## Conclusion

Introducing new concepts comes with obstacles, but proper planning and awareness can make it necessary. Incorporating it into primary healthcare can create new job opportunities and increase familiarity among the general population. Training healthcare professionals on opioid handling can also increase its acceptance.

## Background

Palliative care (PC) represents a new horizon in medical and social sciences [1]. A patient receiving chemotherapy or radiotherapy might also receive palliative care [1, 2]. High-quality palliative care is vital for improving the quality of life for patients and their families. With over 61 million patients requiring this service annually, the demand for palliative care is high globally [3]. The demand is increasing due to changing demographics and illnesses. Sadly, only 14% of those who need it have access to it worldwide [4]. Each year, around 6.5 million people with cancer would need palliative care at the end of their life [5]. Bangladesh ranked second to last in the 2015 EIU QDI report for palliative care [6].

Whereas cancer was the sixth leading cause of death in Bangladesh, with only 4,000 people receiving treatment [7]. Every year, about a million people die in Bangladesh, and an estimated 60 lakh require palliative care [8]. In Bangladesh, cancer-related deaths were at 7.5% in 2005 and are predicted to increase to 13% by 2030. Unfortunately, only 10% of physicians are trained to treat cancer patients, and nearly 60% of young adults are unfamiliar with palliative care. More education and resources are needed to build supporting teams for those affected by cancer in Bangladesh [7, 9–11].

Bangladesh's unevenly distributed and under-resourced PC service units pose a growing public health concern by limiting healthcare access, which is now recognized as a basic human right. Identifying barriers to ensure universal accessibility, especially in end-of-life care, has been recognized as a health ethics challenge. These barriers include misconceptions about treatment options, information gaps, insufficient pain management, physician dilemmas in following advanced directives, and difficulties in delivering accurate health status information to aid decision-making [1, 12]. Besides religious and moral arguments, euthanasia's slow progress is also due to controversy in developed countries [11, 12]. Lack of basic training for healthcare providers, inadequate educational sessions, opioid phobia among healthcare professionals and society, insufficient accessibility, availability, and affordability of opioids, regulatory directions for opioid prescribing, dispensing, and usage, inadequate resources and manpower, unsuitable policies, the escalating burden of life-limiting, non-curable illnesses are all recognized as challenges in different parts of the world [1, 11, 13, 14]. In Bangladesh, a PC team supports patients and families with physicians, nurses, support workers, paramedics, pharmacists, physiotherapists, and community-based health workers (PCA) [15]. Bangladesh has a high

poverty burden, and 2.2 million impoverished people depend on PCAs in informal settings for treatment. They often face challenges with team-building for patient care due to mismanagement. Establishing this sector was impossible without community involvement due to the hospice concept's unfamiliarity in this region [1, 12–17]. So, it is important to address the challenges faced in palliative care units and take measures to overcome them. While many studies focus on the challenges faced by physicians, this study aims to explore obstacles faced by all healthcare providers involved in patient care as a team.

## Methods

### Study design and settings, sample size, and criteria

The study conducted was a descriptive cross-sectional study. The sample was selected using the census method, and 160 participants willingly took part in the study, although the estimated staff number was 178. The data was collected between 1st July 2022 to 30th September 2022. The study population consisted of licensed healthcare providers of both sexes, including doctors, nurses, palliative care assistants, ward staff, and home-based program staff who had been actively involved in providing palliative care to patients for at least one year at BSMMU, Dhaka Medical College & Hospital, Delta Medical College & Hospital, and National Institute of Cancer Research Hospital in Dhaka city, Bangladesh. In addition to these hospital settings, data was also collected from community-based palliative care projects of BSMMU in collaboration with WHPCA in Korail and Narayanganj City Corporation. Physically and mentally unfit providers were excluded from the study. Data was checked in the field immediately after the interview and information gathering.

### Data collection process and analysis

The study used face-to-face semi-structured interviews based on specific variables. The questionnaire was developed in English, and translated into Bangla, and interviews were conducted privately with voluntary participation. Data processing involved categorizing, coding, summarizing, and entering data with SPSS software. Each interview lasted approximately 30 minutes. Categorical and numerical variables were categorized separately. Each "No" answer was given a score of "0," and each "Yes" answer was given a score of "1." Some answers had the option "Unsure" or "Undecided," which were given a score of "2." Descriptive statistics were done for qualitative and quantitative variables, and inferential statistics were used to establish relationships among variables.

### Ethical considerations

The director of the hospitals where the study was conducted gave verbal and written permission, and ethical approval for the research was obtained from the Institutional Review Board (IRB) of NIPSOM. Necessary modifications were made based on their suggestions (Memo no: NIPSOM/IRB/2017/09). Interested participants were required to provide informed written consent before proceeding to the next steps of the questionnaire. The confidentiality of respondents was strictly protected. The investigators were solely responsible for the privacy and confidentiality of personal information, and the data was stored safely. Transfer of data was minimized and not shared with anyone who was not involved in the study.

## Results

In total, 160 responses were received, completed, and analyzed, resulting in a response rate of 89.9%. The participants' baseline sociodemographic characteristics are shown in Table 1. The

**Table 1. Distribution according to socio-demographic traits.**

| Age range | Frequency | Percent |
|---|---|---|
| 20–39 years | 106 | 60.9 |
| 40–59 years | 54 | 39.1 |
| Mean age | 33.59 ± 8.050 | |
| *Sex* | | |
| Male | 51 | 31.9 |
| Female | 109 | 68.1 |
| *Designation* | | |
| Senior doctors (Asst. prof/Assoc. prof/Professor) | 5 | 3.1 |
| Junior doctors (Medical officer/Resident doctor) | 38 | 23.8 |
| SSN | 60 | 37.5 |
| PCA/Ward staff + Junior Staff | 34 | 21.2 |
| Admin staff | 23 | 14.4 |
| Junior staff | 13 | 8.1 |
| *Working experience in the designated post* | | |
| 1-3year | 53 | 33.1 |
| 4-6year | 55 | 67.5 |
| 7-9year | 24 | 82.5 |
| 10-above | 28 | 17.5 |
| *Overall satisfaction about service* | | |
| Dissatisfactory | 30 | 18.8 |
| Satisfactory | 101 | 63.1 |
| Very satisfactory | 29 | 18.1 |
| *Income* | | |
| <20,000 | 48 | 30 |
| 20,000–40,000 | 39 | 24.4 |
| 41,000–60,000 | 20 | 12.5 |
| 61,000–80,000 | 20 | 12.5 |
| 81,000-1lakh (BDT) | 33 | 20.6 |

mean age was 33.59 ± 8.05 years, with over two-thirds being female and only half having tertiary education (52.5%). The study divided the PC team into two broad groups: proficient staff (physicians and nurses) comprising 64.4% of the team, and non-proficient staff (PCA-WS and administrative staff) comprising 35.6%. In terms of designation, only 37.5% of respondents were senior staff nurses while more than one-fourth (26.9%) were physicians. The majority (82.5%) served for 7–9 years. The survey results showed that 99.4% of the participants received training in basic palliative care and management. Only 3.1% of physicians had seniority, while 2.5% had completed a specialization in this field. About 63.1% of the respondents expressed satisfaction with the services provided, and only 20.6% reported earning more than 80,000 taka (BDT) per month. Late-stage cancer (99.4%) was the predominant disease treated in PCU facilities. Almost 60% of patients sought treatment for geriatric diseases, while 36.3% and 35.6% of patients were attended for end-stage renal disease, stroke, and neurological disorders respectively in the PCU. Only 28.1% of patients sought treatment for end-stage cardiovascular disease. In the case of children, cerebral palsy (24.4%) was the second most common condition, with most patients being found in community settings. Pain was the most frequently reported symptom (100%), while hemorrhage was reported the least (31.3%). Respondents also mentioned breathlessness (62.5%), constipation (54.4%), disorientation (51.3%), fatigue (48.1%), and wound infection (33.8%) (S1 Table). The challenges experienced by the

**Table 2. Distribution of respondents according to various forms of challenges.**

| At the hospital level | | | | |
|---|---|---|---|---|
| **Variables** | **Doctor** | **Nurse** | **PCA—WS** | **Admin** |
| Limited OPD facility | 21(41.2) | 32(49.2) | 15(62.5) | 5(62.5) |
| Limited IPD facility | 19(37.3) | 30(46.2) | 18(75) | 6(75) |
| Ethical dilemma | 18(35.3) | 42(64.6) | 17(70.8) | 4(50) |
| Limited dispensable medicine for patients (Free) | 25(49) | 38(58.5) | 20(83.3) | 1(12.5) |
| Lack of telemedicine facility | 26(51) | 27(41.5) | 7(29.2) | 4(50) |
| *While dealing with patient* | | | | |
| Patients' depression regarding clinical outcome | 19(37.3) | 27(41.5) | 12(57) | 2(25) |
| Communication failure with healthcare providers | 32(62.7) | 38(58.5) | 12(57) | 3(37.5) |
| Severely ill or unconscious patient handling | 14(27.5) | 27(41.5) | 15(62.5) | 2(25) |
| Lack of faith in treatment facility | 8(15.7) | 14(21.5) | 13(54.2) | 2(25) |
| Patients' agitation | 26(51) | 20(30.8) | 6(25) | 1(12.5) |
| Difficulty in understanding complex psycho-pathological changes in patient's life | 21(41.2) | 38(58.5) | 17(70.8) | 5(62.5) |
| *Dealing with family* | | | | |
| Overconcern of family members about the patient | 24(47.1) | 31(47.7) | 5(20.8) | 2(25) |
| Ignorance of family members about the patient | 22(43.1) | 26(40) | 17(70.8) | 3(37.5) |
| Persistent demand for institutional care | 18(35.3) | 29(44.6) | 13(54.2) | 2(25) |
| Excessive medicine demand by family | 45.1(23) | 27(41.5) | 18(75) | 4(50) |
| Irrational interference of family in treatment | 9(17.6) | 38(58.5) | 13(54.2) | 4(50) |
| Overwhelmed attitude | 29(56.9) | 35(53.8) | 10(32.5) | 5(62.5) |
| Non-cooperation with the PC team in patient care | 18(35.3) | 28(43.1) | 12(57) | 3(37.5) |
| Providing misleading information | 26(54.1) | 38(58.5) | 12(57) | 4(50) |
| *Opioid use and prescription-related challenge* | | | | |
| Opiophobia seen in other HPs | 38(73.1) | 43(67.2) | 21(58.3) | 6(75) |
| Limited dose formulation | 16(30.8) | 16(34.4) | 15(41.7) | 4(50) |
| Opioid prescribing restriction | 13(25) | 20(31.3) | 15(41.7) | 1(12.5) |
| Social taboo related to opioid use | 26(50) | 37(57.8) | 19(52.8) | 5(62.5) |
| *Administrative level* | | | | |
| Lack of trained staff | 28(54.9) | 4(6.2) | 5(24.1) | 3(37.5) |
| Lack of quality improvement strategy | 15(29.4) | 19(29.2) | 11(45) | 6(75) |
| Lack of enough beds in the facility | 23(45.1) | 28(43.1) | 9(37.5) | 4(50) |
| Admission restriction in home-based care | 18(35.3) | 34(52.3) | 5(24.1) | 5(62.5) |
| *Challenges at-home care* | | | | |
| *a) Institutional level* | | | | |
| Limited supporting staff | 24(47.1) | 60(92.3) | 28(79.5) | 5(62.5) |
| Limited co-operation with NGO | 10(19.6) | 36(55.4) | 30(83.33) | - |
| Lack of specialized training | 29(56.9) | 38(58.5) | 5(24.1) | 3(37.5) |
| *b) Personal level Challenge* | | | | |
| Fear of social discrimination | 37(72.5) | 32(49.2) | 21(58.33) | 4(50) |
| Lack of confidence | 10(19.6) | 28(43.1) | 17(47) | 3(37.5) |
| Job insecurity | 11(21.6) | 2(3.1) | 21(58.33) | 4(50) |
| Lack of career development opportunity | 18(35.3) | 20(30.8) | 20(83.3) | 5(62.5) |
| Cultural conflict | 27(51.9) | 25(39.1) | 13(36.1) | 4(50) |

respondents are presented in Table 2, while their significance is shown in Tables 3 and 4. The challenges vary mainly with the working level, experience, and types of hospitals they work in. At the hospital level, more than half of the physicians (51%) agreed that the unavailability of telemedicine was challenging (p<0.010, p<0.001). It was found that 83.3% of PCA-WS and

**Table 3. Challenges faced at the professional level and hospital type (only yes value is included).**

| Different categories of challenge | Doctor | Nurse | WS/ PCA | Admin and other office staff | P value |
|---|---|---|---|---|---|
| **Ethical dilemma in treatment option** | 19(44.2) | 41(68.3) | 25(73.5) | 4(50) | 0.005* |
| **Lack of telemedicine facility** | 25(49) | 38(58.5) | 9(26.5) | 4(50) | 0.010** |
| **Difficulty in dealing with agitated patients** | 14(27.5) | 30(50) | 26(76.5) | 2(25) | 0.001** |
| **Overconcern of the family** | 25(49) | 33(55) | 5(14.7) | 2(25) | 0.001** |
| **Job insecurity** | 16(37.2) | 37(61.7) | 24(70.6) | 4(50) | 0.003 * |
| **Fear of social discrimination** | 14(32.6) | 32(53.33) | 21(61.8) | 4(50) | 0.01** |
| **Lack of career development opportunity** | 10(23.3) | 28(46.7) | 17(50) | 3(37.5) | 0.01** |
| Hospital types | **Medical College** | **Specialized hospital** | **Super specialized hospital** | | **P value** |
| | | | **Hospital-based** | **Home-based** | |
| **Limited medicine dispensing facility** | 25(52.1) | 18(50) | 32(60.4) | 19(82.9) | 0.049* |
| **Lack of telemedicine facility** | 21(43.8) | 28(77.8) | 19(35.8) | 8(34.8) | 0.001* |
| **Limited co-operation with NGO** | 29(60.41) | 15(41.66) | 13(24.5) | 19(82.6) | 0.001* |
| **Lack of specialization opportunity** | 14(29.2) | 17(47.2) | 12(22.6) | 15(65.2) | 0.002* |
| **Lack of trained administrative staff** | 40(83.3) | 21(58.3) | 42(79.2) | 17(73.9) | 0.021* |

almost 60% of SSN voted in favor of the repeated demand for limited free dispensable medicine, with a significance level of $p < 0.049$. Additionally, 75% of administrative staff identified limited IPD as a challenge. A majority of SSNs (64.6%) recognized ethical dilemmas as a barrier, with a significance level of $p < 0.005$. Proficient respondents (62.7% and 58.5% respectively) identified communication failure as the main challenge when dealing with patients, while non-proficient participants (70.8% and 62.5%) faced difficulties understanding patients' complex psychology. Respondents' experience correlated with their ability to manage challenging psychology ($p < 0.019$).

According to survey conducted, 56.9% of doctors and 62. 5% of administrative staff mentioned that the overwhelmed attitude of family members had made the situation difficult. Around 60% of SSNs had encountered irrational interference from family members while treating patients, which was seen as a challenge. Three out of four (75%) PCA-WS agreed that excessive demand for medicine was a barrier. Almost half of the proficient (47.1%, 47.7% respectively) respondents reported that the overconcern of patients' families was also creating changes while serving patients. Furthermore, there was a significant relationship between the overconcern of families and the professional level of staff ($p < 0.001$); the same result was found with the experience of the employee ($p < 0.008$). The majority of respondents across all four categories reported that their use and prescription of opioids were hindered by the fear of opioids among other healthcare providers (73.1% of doctors, 67.2% of nurses, 58.3% of WS-PCA, and 75% of administrative staff) (Table 2).

**Table 4. Challenges based on professional working experience (only yes included).**

| Challenges based on work experience | | | | | |
|---|---|---|---|---|---|
| Categories of challenge | 1–3 years | 4–6 years | 7–9 years | 10 years and above | p-value |
| Over concern of family | 31(58.5) | 17(30.9) | 6(25) | 11(39.3) | 0.008* |
| Non-co-operation attitudes of patients | 16(30.2) | 22(40) | 8(33.3) | 17(60.7) | 0.024* |
| Difficulty in understanding complex psycho-pathological changes of patient's life | 24(45.3) | 33(60) | 8(33.3) | 20(71.4) | 0.019* |
| Difficulty in overcoming depression following death despite excellent care | 26(49.1) | 38(69.1) | 4(16.7) | 17(60.7) | 0.001* |
| Limited dose formulation of opioids | 31(58.5) | 20(36.4) | 18(75) | 19(67.9) | 0.007* |
| Facing trouble using opioids as a medicine | 17(32.1) | 14(25.5) | 16(66.7) | 10(35.7) | 0.005* |

**Table 5. Satisfaction Level of provided service on performance skill.**

| Service categories | Very dissatisfactory | Dissatisfactory | Undecided | Satisfactory | Very satisfactory |
|---|---|---|---|---|---|
| Pain relief | 11(6.9) | 16(10) | 26(16.3) | 47(29.4) | 60(37.5) |
| Response to Side effect | 22(13.8) | 30(18.8) | 19(11.9) | 56(35) | 33(20.6) |
| Reliving patients concern | 23(14.4) | 42(26.3) | 30(18.8) | 31(19.4) | 34(21.3) |
| Family mitting discussion | 23(14.4) | 46(28.7) | 24(15) | 41(25.6) | 26(16.3) |
| HP's role in choosing right treatment | - | 32(20) | 15(9.4) | 27(16.9) | 86(53.8) |
| Community participation in home-based setup | 47(29.4) | 43(26.9) | 11(6.9) | 20(12.5) | 39(24.4) |
| Pain management performance | 28(17.5) | 12(7.5) | 21(13.1) | 54(33.8) | 45(28.1) |
| Opioid utilization performance | 13(8.1) | 32(20) | 53(33.1) | 51(31.9) | 11(6.9) |
| Existing training strategies are | 6(3.8) | 39(24.4) | 55(34.4) | 46(28.7) | 14(8.8) |
| HPs participation in decision-making | - | 14(8.8) | 44(27.5) | 70(43.8) | 32(20) |
| HPs knowledge level for dealing patient | 1(0.6) | 42(26.3) | 50(31.3) | 61(38.1) | 6(3.8) |
| Availability of treatment equipment in setup | 14(8.8) | 52(32.5) | 46(28.7) | 48(30) | - |
| Availability of opioids and supportive drugs | 4(2.5) | 34(21.3) | 51(31.9) | 57(35.6) | 14(8.8) |

However, more than one-third (31.87%) of participants from all four groups claimed that the burden of opioid use was influenced by the narrow dose formulation, and there was also a significant relationship found with work experience (p<0.007). At the administrative level, nearly 55% of doctors agreed that the lack of trained staff was a challenge. Additionally, 75% of administrative staff and 45% of PCA-WS identified the lack of quality improvement strategies as a challenge. Home care challenges have been divided into two main categories: institution-oriented and personal level. Institution-oriented challenges include limited supporting staff, limited NGO cooperation, and a lack of specialized training facilities. Almost all SSNs (92.3%), 62.5% of administrative staff, and nearly half of the doctors (47.1%) identified limited supporting staff as a major problem for home care. There was a significant relationship (p<0.021) found with the type of hospital providing the service (**Table 3**).

At a personal level, individuals face various challenges such as fear of social discrimination, lack of confidence, job insecurity, lack of career development opportunities, and cultural conflicts. In the study, it was found that 66.99% of proficient participants recognized fear of social discrimination as the major personal challenge and a significant relationship was found with the professional level (p<0.010). On the other hand, lack of career development opportunities was identified as a vital personal level challenge by 83.3% and 62.5% of PCA-WS and AS non-proficient groups, respectively (p<0.010) (**Tables 4 and 5**) and (**S2 Table**).

## Discussion

This cross-sectional study aimed to explore barriers faced by healthcare providers in hospitals and home-based palliative care units. The study discovered that the difficulties faced by employees vary depending on the type of work environment, designation & experience. For better narrative flow, challenges were categorized into workplace-related challenges (hospital level, administrative level, opioid use related, and homecare) and subjective-oriented challenges (personal level, patients and their families). The HPs encountered several barriers that hindered their ability to provide adequate patient care in the hospital. Limited outpatient and inpatient services, insufficient availability of cost-free necessary medication, and the absence of telemedicine facilities, ethical dilemmas were the main subcategories in this section. In PCU, healthcare professionals face ethical dilemmas that further complicate their role in delivering patient care. In the study, nearly two-thirds of healthcare professionals agreed that ethical dilemma and their role were major obstacles in providing patient care. They often hesitate to

reveal the actual condition and offer treatment options, considering the patient's fitness and financial backup. Additionally, societal acceptance of palliative care is lower compared to other curative options. In Bangladesh, discussing death is considered taboo and families refuse to accept negative medical outcomes, regardless of financial means. Healthcare providers (HPs) may feel uncertain and perplexed when treating terminal or end-of-life patients. Published studies suggest that doctors hesitate to perform CPR, DNR, or suggest alternative options to patients due to social and cultural practices [8, 18–21]. Chiu et al. (2000) identified ethical dilemmas such as truth-telling, place of care, therapeutic strategy, hydration and nutrition, blood transfusion, alternative treatment, terminal sedation, and medication use in symptom management. Physicians face ethical dilemmas when referring patients, as they must predict a patient's 6-month prognosis accurately. Clinicians must provide complete information about a patient's illness, as required by the Medicare Hospice Benefit [18]. With proper awareness and spiritual education, it is possible to eliminate these negative emotions. Despite a decade-long journey, Bangladesh's focus remains limited to relieving pain to improve patients' quality of life, while ignoring their sufferings due to exclusion from primary healthcare [11]. Open seminars on patient management for palliative care could be a useful solution to reach a large number of people. While providing care for patients and their family members, healthcare professionals face various challenges. These challenges can be broadly categorized into two sub-groups. The first group includes challenges that arise while handling patients, such as depression related to their clinical outcome, communication breakdowns, dealing with severely ill or unconscious patients, patient agitation, and difficulty in understanding complex psychological changes in the patient's life. In the study, it was found that communication failure was a common issue faced by all groups involved. Patients often experience a sense of unimportance and detachment from their personality when diagnosed with a life-limiting illness. They may deny the true clinical outcome despite the evidence. As a result, social isolation is commonly observed among them, making it difficult for healthcare professionals to establish a strong and trustworthy bond with them, and bringing renewed hope may seem impossible. Nurses commonly encounter challenges in building a connection with patients who hold cultural beliefs that conflict with modern medicine, as noted by Ferrel et al. in 2006 [22]. Occasionally, the traditional spiritual beliefs of terminally ill patients prove to be of little comfort, as they fervently hope for a miracle despite their poor prognosis. Some patients may even refuse to acknowledge their illness and may be unwilling to communicate with healthcare providers in the PCU [23, 24]. During the process of dealing with family, healthcare providers may face obstacles such as excessive demands for medicine, overconcern of family non-cooperation with the PC team, overwhelmed attitudes of family, and providing misleading information about the patient. Many doctors and other healthcare professionals are frequently faced with overwhelmed patients and often receive misleading information. Additionally, an excessive amount of concern can also worsen the situation. Families may feel helpless when there are no curable options available, while also trying to accept new concepts of treatment. They may think that using large doses of pain medicine may reduce their loved one's suffering, but this can sometimes misguide healthcare professionals. According to Waldrop (2007), the most common psychological and emotional responses to grief are intense sadness and anger. Family members, who are often angry and anxious, may refuse doctors' instructions, even if they do not consider the feelings of the nurses [24]. Families often ignore doctors' warnings and choose aggressive end-of-life treatments, fueled by unrealistic optimism about patient outcomes [25]. According to Beckstrand et al. (2009), supportive behavior is essential in end-of-life care. Such behavior includes spending time with the patient's family during the terminal stage, creating a private corner for them, having healthcare providers (HPs) present, and reducing the patient's discomfort. Additionally, an open discussion session with both the

patient and family can help them overcome any challenges they may face during this difficult time. A significant number of respondents have acknowledged the shortage of palliative essential medications such as opioids. Opioids are part of palliative patient management not only for pain relief but also for reducing patients' agitation in the terminal stage [14]. In Bangladesh, opioids are not included in the essential drug list, which means that the PCU has had to modify its strategies at times. It may happen because the country has poor handling and monitoring systems. Besides, there is a significant possibility of developing addiction or tolerance if not used meticulously. In the study opioid phobia among other health practitioners was identified as a prime challenge by the majority of respondents. Professionals working other than PCU are not familiar with opioid use in their practice. The use of certain medications may cause anxiety among healthcare professionals and caregivers as these medications could potentially violate public health regulations [13]. Evidence from published studies shows that the consumption rate of opioids was only 5 grams per capita in 2014. Additionally, in Bangladesh, opioids, particularly morphine, are only available in syrup and tablet forms and are expensive compared to the developed world. The limited use of opioids despite their good outcomes may be due to a lack of knowledge about their accessibility, availability, and use [23, 26, 27]. Besides, there is no official training session regarding opioid use and side effect management at the policy level. However, the government has already taken the initiative to enhance knowledge about PC and opioid use among doctors and nurses due to the rising burden of cancer for appropriate use feasible restriction policy is needed unless there might be a chance to develop addiction, dependence, and tolerance [13, 14]. Developing transdermal patches can enhance popularity among doctors and eradicate opioid phobias. In the study, more than half of the respondents from all groups saw this as a challenge. To avoid hazardous incidents, the PCU professionals followed prescription restrictions, but when the burden of patients was high about the service provider, it became increasingly difficult. A study by Mbozi et al. found that nurses were hindered from providing adequate pain management in PCU due to restrictions on their ability to prescribe morphine [14]. On the contrary, according to the study, respondents who were not doctors strongly supported the idea that prescription rights should be reserved for registered doctors. However, they also believe that there is a need for expansion of prescription authority, which can be achieved through proper policy and vocational training. Additionally, it is important to elaborate on the concept of pain management and focus on other aspects of pain management among health professionals, which can potentially reduce these problems. WHO defines palliative care as a basic human right aimed at reducing the suffering of patients and their families in all aspects [13]. China and India have established advanced local community-based systems compared to the West [5]. In developed countries, palliative care is typically integrated into the primary healthcare system. HP adopts policies, strategies, and training sessions to enhance skills in handling complex biomedical and psychological issues [17]. Despite the significant patient burden, access to palliative care remains limited [5, 13]. Palliative care is still in the developing phase, which means that policies and strategies are not fully developed yet. At the administrative level, there are frequent notices of a lack of beds, quality improvement policies, and admission restrictions in home-based systems. The barriers vary significantly for skillful and partially skillful healthcare professionals, depending on their experience. In addition, creating a skillful, community-oriented team requires robust professional and financial support [1, 13–17]. In Bangladesh, not all PCUs offer home-based services. There are a variety of reasons for this, which can be divided into two categories. Firstly, institutional reasons include limited supporting staff, lack of cooperation with local NGOs, and insufficient professional training for the respondents. Almost all nursing staff and nearly half of the doctors reported that they have limited staff at the service facility. More than half of the nursing staff and almost a quarter of the physicians stated that

the lack of cooperation from community-based NGOs was a barrier to running community-based programs. Secondly, there were also personal barriers for the respondents, which were further divided into the fear of social discrimination, job insecurity, lack of confidence in handling patients and their families outside the hospital, and lack of career development opportunities. Many healthcare professionals did not view their jobs as prestigious due to the fact they were often required to work in fields that were not considered as such. As a result, nearly 80% of doctors, 50% of nurses, 60% of PCA-WS, and 50% of AS were afraid of experiencing social discrimination. These negative attitudes ultimately led to feelings of demotivation among healthcare professionals. Approximately one-fifth of doctors, half of the nurses, and PCA-WS stated that at times, they were unable to provide highly individual specialized service which often resulted in anger from the patient's family. To provide proper care, staff had to frequently visit the patient. However, they felt depressed and less confident while dealing with them. Furthermore, employees who work in home care tend to feel uncertain and less contented about their career growth prospects and the available amenities. A similar scenario was observed in research conducted by Zhang and Chiu in China, where nurses serving patients in home-based settings were less motivated [18, 19]. This was because patients' family members considered them to have a lower status in comparison to their colleagues working in hospitals [18, 19, 24, 28–31]. Another study by Rahman and Ferdous stated that healthcare professionals working at PCUs in Bangladesh have limited opportunities to enhance their professional skills due to the lack of facilities [8, 18]. Several studies have highlighted the differences in facilities, career progression, and training opportunities between hospital-based and home-based staff. These differences can be addressed by integrating palliative home-based services into the national primary healthcare system. Advanced training programs and collaboration with cross-border Palliative Care Units (PCUs) can enhance the patient handling capacity and employee confidence. The studies have also shown that job satisfaction and knowledge positively impact service providers, leading to reduced conflicts and improved service quality [8, 18, 28]. The study revealed that 63.1% of respondents view the lack of home-based palliative care (HBPC) services as a significant challenge. Currently, the super-specialized hospital, BSMMU, is the sole provider of home-based services in Bangladesh, resulting in cost savings for the healthcare system. In a 2019 study conducted by Biswas et al., it was emphasized that home-based palliative care (HBPC) services play a crucial role in improving the quality of life (QOL) for terminally ill patients [22] Similar positive effects were also observed in various regions around the world, regardless of the patients' access to family and psychological support systems [22, 25, 32–37]. However, many other centers encounter challenges such as limited manpower and financial constraints, inadequate home-oriented guidelines. In present HC, patients are enrolled based on preset (PPS below 50, distance, absence of caregiver) criteria and receive 2–3 visits per month from healthcare professionals on need basis. The study also agrees that doctor visits are infrequent, telemedicine services are available but inadequate as it fail to enhance HP' s work satisfaction [22, 25, 32–34, 38]. Nevertheless, almost two-thirds of participants expressed dissatisfaction with the provided services due to the lack of practical home-oriented guidelines (68.1%) and an effective telemedicine service (47.7%) system. Research in Dhaka, Bangladesh, indicated that existing home-based care services fail to effectively manage symptoms in over half of the patients, underscoring the necessity for proper treatment protocols and improved communication [33]. Different published studies agreed with the notion that home care for elderly and non-self-sufficient patients is vital in addressing the growing demand for health services due to the aging population and reducing the strain on hospitals [33, 39–42]. Multi-professional home-based palliative care teams, comprising doctors, nurses, physiotherapists, psychologists, and healthcare assistants, are crucial for addressing the complex health and social needs of patients also compatible with other published

studies [22, 40]. Additionally, the importance of monitoring patients digitally, both in healthcare facilities and at home, has become more significant, particularly in the wake of the pandemic. Digital applications play a vital role in enhancing patient compliance and improving cost efficiency [41, 43]. Furthermore, digital applications can aid individuals with disabilities, ensuring their full participation in social, relational, and work activities. However, developing a user-friendly digital system is time-consuming, costly, and may breach user privacy, so feasibility testing is necessary depending on societal context [40, 42, 44–46].

## Limitations

Since the study design was descriptive and cross-sectional, respondents only provided their responses at a single point in time. The challenges faced by these respondents were subjective and varied from person to person. The study was limited to larger institutional and home-based facilities and excluded different hospice-based setups. Therefore, the findings cannot be generalized to all settings. Additionally, due to the limited availability of home-based setups to provide palliative care, less focus was given to that section.

## Recommendations

To overcome the challenges, it's crucial to formulate realistic and feasible policies tailored to the needs of each country. PCU's expansion is necessary for it to yield fruitful results. An appropriate referral system can help to minimize the distance between doctors of different departments and PCU, improving patient care. Conferences and training sessions are effective ways to reduce opioid phobia among healthcare professionals. Moreover, integrating PCU into the primary healthcare system enhances acceptance by patients and their family members and increases employee satisfaction by reducing fear of social discrimination.

## Conclusion

Palliative care has been around for over a decade, yet it still does not receive the recognition it deserves. It is not only meant for patients at the end of their life. By providing adequate vocational training and awareness programs, knowledge can be built.

Raising awareness among palliative care providers is vital for improving the quality and efficiency of care. Strategies include workshops, continuing medical education, online platforms, multidisciplinary team meetings, mentorship programs, and peer support groups. Access to guidelines, telemedicine, mobile apps, and regular feedback mechanisms are also important. Additionally, community engagement and policy advocacy are needed to promote palliative care education and resources.

## Supporting information

**S1 Table. Distribution as mostly handled patients & predominant symptoms (n = 160).** (DOCX)

**S2 Table. Relationship between professional level and satisfaction of HP (n = 160).** (DOCX)

## Acknowledgments

The authors gratefully acknowledge the contribution of the Director of Dhaka Medical College, BSMMU, DLMC & NIRCH, and home care team members of BSMMU and their coordinator to provide information during data collection.

## Author Contributions

**Conceptualization:** Mastura Kashmeeri, A. N. M. Shamsul Islam.

**Data curation:** Mastura Kashmeeri, Palash Chandra Banik.

**Formal analysis:** Mastura Kashmeeri.

**Funding acquisition:** Mastura Kashmeeri.

**Investigation:** Mastura Kashmeeri.

**Methodology:** Mastura Kashmeeri.

**Project administration:** Mastura Kashmeeri.

**Resources:** Mastura Kashmeeri.

**Software:** Mastura Kashmeeri.

**Supervision:** Mastura Kashmeeri.

**Validation:** Mastura Kashmeeri, A. N. M. Shamsul Islam, Palash Chandra Banik.

**Visualization:** Mastura Kashmeeri, A. N. M. Shamsul Islam, Palash Chandra Banik.

**Writing – original draft:** Mastura Kashmeeri, A. N. M. Shamsul Islam.

**Writing – review & editing:** Mastura Kashmeeri, A. N. M. Shamsul Islam, Palash Chandra Banik.

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
