## [Decision Letter · Decision Letter 0]

11 Jun 2024

PONE-D-24-15132Date: April 8, 2024

Challenges experienced by health care providers working in both hospital and home-based palliative care units in Dhaka city: A multi-center based cross-sectional studyPLOS ONE

Dear Dr. Kashmeeri,

Thank you for submitting your manuscript to PLOS ONE. After careful consideration, we feel that it has merit but does not fully meet PLOS ONE’s publication criteria as it currently stands. Therefore, we invite you to submit a revised version of the manuscript that addresses the points raised during the review process.

We look forward to receiving your revised manuscript.

Kind regards,

Roberto Scendoni

Academic Editor

PLOS ONE

2. Please provide additional details regarding participant consent. In the ethics statement in the Methods and online submission information, please ensure that you have specified (1) whether consent was informed and (2) what type you obtained (for instance, written or verbal, and if verbal, how it was documented and witnessed).

“BMRC, Fund no 73”

“none”

5. Please amend either the title on the online submission form (via Edit Submission) or the title in the manuscript so that they are identical.

Additional Editor Comments:

The authors, in the conclusion section, should better explain how to raise awareness among healthcare professionals working in the Palliative Care Unit with the concepts introduced. Ex: subject them to periodic questions? Carry out training courses? Ensure internal audits? Implement contact with patients? Please discuss these aspects further.

Please expand the number of references regarding telemedicine and home care as suggested by the reviewer.

Reviewers' comments:

Reviewer's Responses to Questions

**Comments to the Author**

1. Is the manuscript technically sound, and do the data support the conclusions?

Reviewer #1: Yes

Reviewer #2: Yes

2. Has the statistical analysis been performed appropriately and rigorously? 

Reviewer #1: Yes

Reviewer #2: Yes

3. Have the authors made all data underlying the findings in their manuscript fully available?

Reviewer #1: Yes

Reviewer #2: Yes

4. Is the manuscript presented in an intelligible fashion and written in standard English?

Reviewer #1: Yes

Reviewer #2: Yes

5. Review Comments to the Author

Reviewer #1: The study is well structured.

The data are clearly reported.

The various sections are written comprehensively and plainly.

However, I feel that the authors need to better explore the topics of telemedicine and homecare better. There are numerous papers, including recent ones such as 10.3389/fpubh.2022.1095001, regarding these issues.

Reviewer #2: Palliative care is paramount in the modern clinical field worldwide. This study aimed to explore the challenges faced by healthcare providers working in the palliative care unit in Bangladesh. The authors of the article have brought new and recent information that can be of use at the local level in the country. The article is prepared precisely and can be accepted for publication.

6. PLOS authors have the option to publish the peer review history of their article (what does this mean?). If published, this will include your full peer review and any attached files.

Reviewer #1: No

Reviewer #2: No

---

## [Author Response · Author response to Decision Letter 0]

14 Jun 2024

Response to reviewer

Manuscript title Challenges experienced by health care providers working in both hospital and home-based palliative care units in Dhaka city: A multi-center based cross-sectional study

Manuscript ID PONE-D-24-15132

Reviewer 1’s comments to the authors

Comment1 :

The study is well structured.

The data are clearly reported.

The various sections are written comprehensively and plainly.

However, I feel that the authors need to better explore the topics of telemedicine and homecare better. There are numerous papers, including recent ones such as 10.3389/fpubh.2022.1095001, regarding these issues. 

Reply: Thank you for your suggestion. We have added your suggested information in the discussion. 

Line numbers 347 to 375 (From clean copy)

Changes in the text

The study revealed that 63.1% of respondents view the lack of home-based palliative care (HBPC) services as a significant challenge. Currently, the super-specialized hospital, BSMMU, is the sole provider of home-based services in Bangladesh, resulting in cost savings for the healthcare system. In a 2019 study conducted by Biswas et al., it was emphasized that home-based palliative care (HBPC) services play a crucial role in improving the quality of life (QOL) for terminally ill patients.[33] Similar positive effects were also observed in various regions around the world, regardless of the patients' access to family and psychological support systems. [33-35, 37-39]. However, many other centers encounter challenges such as limited manpower and financial constraints, inadequate home-oriented guidelines. In present HC, patients are enrolled based on preset (PPS below 50, distance, absence of caregiver) criteria and receive 2-3 visits per month from healthcare professionals on need basis. The study also agrees that doctor visits are infrequent, telemedicine services are available but inadequate as it fail to enhance HP’ s work satisfaction. [33-36]. Nevertheless, almost two-thirds of participants expressed dissatisfaction with the provided services due to the lack of practical home-oriented guidelines (68.1%) and an effective telemedicine service (47.7%) system. Research in Dhaka, Bangladesh, indicated that existing home-based care services fail to effectively manage symptoms in over half of the patients, underscoring the necessity for proper treatment protocols and improved communication. Different published studies agreed with the notion that home care for elderly and non-self-sufficient patients is vital in addressing the growing demand for health services due to the aging population and reducing the strain on hospitals. [40-41,43-44]. Multi-professional home-based palliative care teams, comprising doctors, nurses, physiotherapists, psychologists, and healthcare assistants, are crucial for addressing the complex health and social needs of patients also compatible with other published studies. [33, 41]. Additionally, the importance of monitoring patients digitally, both in healthcare facilities and at home, has become more significant, particularly in the wake of the pandemic. Digital applications play a vital role in enhancing patient compliance and improving cost efficiency. [42,43]. Furthermore, digital applications can aid individuals with disabilities, ensuring their full participation in social, relational, and work activities. However, developing a user-friendly digital system is time-consuming, costly, and may breach user privacy, so feasibility testing is necessary depending on societal context. [41, 44-47].

Reviewer 2 ’s comments to the authors

Comment 1 :

Reviewer #2: Palliative care is paramount in the modern clinical field worldwide. This study aimed to explore the challenges faced by healthcare providers working in the palliative care unit in Bangladesh. The authors of the article have brought new and recent information that can be of use at the local level in the country. The article is prepared precisely and can be accepted for publication.

Reply: Thank you for your compliment we feel greatly honored.

Editor’s comments to the authors

The authors, in the conclusion section, should better explain how to raise awareness among healthcare professionals working in the Palliative Care Unit with the concepts introduced. Ex: subject them to periodic questions? Carry out training courses? Ensure internal audits? Implement contact with patients? Please discuss these aspects further.

Please expand the number of references regarding telemedicine and home care as suggested by the reviewer.

Reply: Thank you for your suggestion. We have added your required information in the conclusion

Line numbers 392 to 400 (From clean copy)

Changes in the text

Palliative care has been around for over a decade, yet it still does not receive the recognition it deserves. It is not only meant for patients at the end of their life. By providing adequate vocational training and awareness programs, knowledge can be built. 

Raising awareness among palliative care providers is vital for improving the quality and efficiency of care. Strategies include workshops, continuing medical education, online platforms, multidisciplinary team meetings, mentorship programs, and peer support groups. Access to guidelines, telemedicine, mobile apps, and regular feedback mechanisms are also important. Additionally, community engagement and policy advocacy are needed to promote palliative care education and resources.

Response to journal requirements:

1. Please ensure that your manuscript meets PLOS ONE's style requirements, including those for file naming. The PLOS ONE style templates 

Reply: Thank you for the recommendation, changes have been made as per requirements.

2. Please provide additional details regarding participant consent. In the ethics statement in the Methods and online submission information, please ensure that you have specified (1) whether consent was informed and (2) what type you obtained (for instance, written or verbal, and if verbal, how it was documented and witnessed).

Reply: Thank you for raising the point, changes have been made to the clean copy manuscript ethical consideration.

Line numbers 135 to 136 (From clean copy)

Changes in the text

Interested participants were required to provide informed written consent before proceeding to the next steps of the questionnaire.

Reply: Thank you for raising the point, changes have been made to the clean copy manuscript.

Line numbers 406 to 408 (From clean copy)

Changes in the text

The authors received financial support from the Bangladesh Medical Research Council (BMRC). However, the funders had no involvement in the study design, data collection and analysis, the decision to publish, or the preparation of the manuscript.

Reply: Thank you for raising the point, changes have been made to the clean copy manuscript in the reference section with DOI.

Line numbers 414 to 560 (From clean copy)

---

## [Editor Report · Decision Letter 1]

24 Jun 2024

Challenges experienced by health care providers working in both hospital and home-based palliative care units in Dhaka city: A multi-center based cross-sectional study

PONE-D-24-15132R1

Dear Dr. Kashmeeri,

We’re pleased to inform you that your manuscript has been judged scientifically suitable for publication and will be formally accepted for publication once it meets all outstanding technical requirements.

Kind regards,

Roberto Scendoni

Academic Editor

PLOS ONE
---

## [Editor Report · Acceptance letter]

8 Jul 2024

PONE-D-24-15132R1 

PLOS ONE

Dear Dr. Kashmeeri, 

I'm pleased to inform you that your manuscript has been deemed suitable for publication in PLOS ONE. Congratulations! Your manuscript is now being handed over to our production team.

Kind regards, 

on behalf of

Dr. Roberto Scendoni 

Academic Editor

PLOS ONE